# Transcultural Adaptation and Evaluation of the Psychometric Properties of the Spanish Version of the FCR7 Questionnaire for Assessing Fear of Recurrence in Cancer Patients: FCR6/7-SP

**DOI:** 10.3390/cancers17050875

**Published:** 2025-03-03

**Authors:** Cristina Díaz-Periánez, Rafaela Camacho-Bejarano, Héctor González-de la Torre, Susanne Cruickshank, Gerald Michael Humphris, Eloísa Bayo-Lozano, Dolores Merino-Navarro

**Affiliations:** 1Department of Nursing, University of Huelva, 21071 Huelva, Spain; cristina.perianez@denf.uhu.es (C.D.-P.); lola.merino@denf.uhu.es (D.M.-N.); 2Nursing and Healthcare Research (Investen-ISCIII), 28029 Madrid, Spain; 3Department of Nursing, University of Las Palmas de Gran Canaria, 35017 Gran Canaria, Spain; hector.gonzalez@ulpgc.es; 4Royal Marsden NHS Foundation Trust, London SW3 6JJ, UK; susanne.cruickshank@rmh.nhs.uk; 5Medical School, University of St Andrews, St Andrews KY16 9TF, UK; gmh4@st-andrews.ac.uk; 6Health Science Doctorate Program, University of Huelva, 21071 Huelva, Spain; eloisabayo@gmail.com; 7Radiotherapy Oncology Unit, University Hospital Virgen Macarena, 41009 Seville, Spain

**Keywords:** oncology nursing, fear of cancer recurrence, validation studies, psychometrics, questionnaires, quality of life

## Abstract

Fear of recurrence is one of the main problems affecting cancer patients once treatment has been completed, and it affects their quality of life. However, there are no tools in Spanish to assess it at the clinical level. The aim of this study is to carry out a cultural adaptation and evaluate the metric properties of the Spanish version of the Fear of Cancer Recurrence (FCR7) questionnaire. After translation and cultural adaptation, the Spanish version is valid and reliable to assess the fear of cancer recurrence, with two models available (FCR7-SP and FCR6-SP). The application of this tool will allow the detection and approach of this phenomenon in clinical practice in cancer survivors.

## 1. Introduction

Cancer represents one of the most significant health challenges in developed countries [1], being one of the leading causes of illness and death, affecting an increasing number of people. As cancer patient survival improves, new emotional concerns arise, especially after treatment completion [2]. Despite advances in cancer treatment, many survivors face the possibility of its recurrence, leading to what is known as “fear of cancer recurrence” (FCR). This was described by Lebel [3] as “fear or worry that cancer might return or progress in the same place or elsewhere in the body,” although its definition is not yet fully agreed upon [4]. This fear significantly limits the quality of life and emotional well-being of affected individuals [5]. Furthermore, it can impact decision-making related to health, lifestyle, and social interactions [6].

Internationally, a systematic review with meta-analysis conducted in 2022 estimated that more than half (59%) of cancer survivors experience at least a moderate level of fear of recurrence, and approximately one in five experiences a high level of fear [5]. Even when the risk of recurrence is low, the fear remains consistently high for years after treatment [4], having a significant impact on patients’ quality of life. FCR can become very distressing, chronic, and disabling, and it is associated with negative health outcomes such as depression, anxiety, and generalized anxiety disorder [7,8]. Additionally, FCR predicts potential alterations in health-related behaviors, such as hypervigilance toward bodily symptoms, increased use of psychotropic medications, greater use of healthcare services, and the utilisation of complementary medicine [8,9,10].

Understanding and assessing fear of recurrence are essential to providing effective support to patients and their families [11]. Early identification of this fear allows for the implementation of management strategies and emotional support, as well as personalized care to address the individual needs of each patient [12]. Despite the importance of assessing fear of recurrence, there is a shortage of specific and validated measurement tools for this population in Spanish. Internationally, several tools have been identified to measure fear of recurrence, some of which are widely used and available in many other languages. One of the challenges observed is that these tools tend to be lengthy, which extends completion time and limits their use in clinical practice. In recent years, a widely disseminated tool has been developed for its versatility, which, unlike previous ones, is shorter and therefore more applicable. For this reason, this instrument has been chosen, as it provides maximum information with optimal data validity in the shortest time possible, and could be of great clinical utility to identify patients at risk of developing this issue and to design effective interventions that can address their needs and prevent possible complications. Therefore, the aim of this study is to adapt the Fear of Cancer Recurrence (FCR7) instrument to the Spanish context and evaluate its psychometric properties.

## 2. Materials and Methods

This research was conducted in two stages: Stage 1 focused on the methodological study and Stage 2 consisted of a cross-sectional study to obtain a validation sample for the evaluation of different psychometric properties of the Spanish version of the FCR7 instrument.

### 2.1. Stage 1: Methodological Study

The translation and transcultural adaptation of the original English version of the FCR7 scale were carried out, along with the evaluation of content validity through expert review and of face validity through a pilot test in the target population.

#### 2.1.1. Starting Instrument

The original version of the FCR7 questionnaire in English was developed at the University of St. Andrews (Scotland) [13] and is used to assess fear of recurrence in cancer patients. This instrument consists of seven questions with a Likert scale from 1 to 5, except for Item 7, which is scored from 1 to 10. The total score ranges from 7 to 40 points, with higher scores indicating a greater fear of recurrence. The reported reliability (internal consistency) for the original scale is good, with a Cronbach’s alpha coefficient of 0.92 (95% CI: 0.90–0.94). Additionally, the initial questionnaire included an abbreviated version, FCR4, with the first four items, designed to facilitate its clinical applicability [13].

#### 2.1.2. Translation Procedure

After obtaining permission from the author of the original questionnaire, the translation process began, carried out by two bilingual translators, one of whom was a native English speaker and familiar with the instrument. The translators provided an adjusted version to ensure comprehension. Subsequently, a back-translation was performed by two Spanish researchers who were unfamiliar with the original questionnaire and produced an English version. These translations were compared to analyze discrepancies until a consensus was reached. A report detailing the identified semantic and conceptual differences was issued, and the final translated version of the questionnaire was presented.

#### 2.1.3. Content Validity

Content validity was assessed through a panel of experts, which consisted of 10 professionals from Medicine and Nursing with the following profiles: 2 oncologists, 1 clinical oncology unit supervisor, 1 advanced practice nurse in complex oncology processes, 3 clinical nurses, and 3 research professors in the field of Nursing (8 women and 2 men). All participants met the established criterion of having specific training/accreditation and/or clinical or research experience in the field of oncology. The experts rated each item according to its relevance (whether the item assessed what it was intended to assess, and the importance of the item in relation to the construct of the study). These criteria were evaluated using a Likert scale ranging from 1 (item not relevant) to 4 (item very relevant). The Content Validity Index for each item (CVI-i) was calculated based on the experts’ ratings. Aiken’s V coefficient, with the respective 95% confidence intervals for each item, was computed [14]. CVI-i values greater than 0.70 were considered adequate [15].

#### 2.1.4. Face Validity

To evaluate face validity, a pilot test of the instrument was conducted with 20 oncology patients with various types of cancer and a profile similar to the target population. All difficulties reported by the participants regarding the wording and comprehensibility of the instrument were collected.

Finally, this version was assessed using the INFLESZ scale [16]. This scale measures the comprehensibility of health texts intended for the general population in the Spanish context. The scale establishes the following classification system: 0 to 40—Very difficult, 40 to 55—Somewhat difficult, 55 to 65—Normal, 65 to 80—Quite easy, and 80 to 100—Very easy. The final Spanish version was named as FCR7-SP and was subjected to a subsequent validation study (Appendix A).

### 2.2. Stage 2: Cross-Sectional Study to Obtain a Validation Sample for the Evaluation of Different Psychometric Properties of the FCR7-SP Instrument

#### 2.2.1. Design

A cross-sectional study to evaluate construct validity, calculate reliability (internal consistency), assess convergent–divergent validity, and validate by known groups for the Spanish version.

#### 2.2.2. Population and Study Setting

The study population consisted of patients diagnosed with cancer who were in the follow-up phase at the Juan Ramón Jiménez University Hospital (HUJRJ), a general hospital specializing in oncology in the province of Huelva (Spain).

Inclusion criteria were as follows: Adults (over 18 years old), capable of reading and understanding Spanish, and diagnosed with non-metastatic cancer under follow-up for at least 6 months after the completion of active treatment. Exclusion criteria were patients with hearing or vision impairments, mental health issues, those receiving palliative care, and patients with altered levels of consciousness.

#### 2.2.3. Sample Size

The Fear of Cancer Recurrence (FCR7) scale was initially designed as a unidimensional instrument consisting of 7 items. According to classical factor analysis theory (FA), at least 10–15 subjects per item of the instrument being validated are required [17]. Based on this and the recommendation to have a minimum of 200 subjects for performing factor analysis using a polychoric correlation matrix [17], an initial minimum sample size of 200 participants was estimated for conducting a factor analysis. Subsequently, an estimation was made using the SENECA method to obtain an empirical estimate of the minimum sample size required for the dataset subject to factor analysis [18].

#### 2.2.4. Variables Under Study

Firstly, sociodemographic variables have been included: date of birth, sex, place of residence, nationality, marital status, composition of the household, number and age of children, family responsibilities, education level, employment status, and income. Clinical variables were also included: type of tumor; date of diagnosis; tumor stage according to the general classification system related to the degree of dissemination, the size and location of the tumor, the involvement of lymph nodes, and the existence of metastases—from stage I (small located tumor) to stage IV (advanced and disseminated cancer); received treatment; date of completion of active treatment; current treatment continuity; visits to a psycho-oncologist; visits to the Comprehensive Oncology Unit at HJRJ; contact with the Advanced Practice Nurse; and use of social support resources. Dependent variables included the fear of recurrence, resilience, and coping strategies.

#### 2.2.5. Instruments and Data Collection System

Data collection was conducted from June 2023 to February 2024. Participants were recruited through consecutive non-probability sampling during follow-up consultations at the Oncology Department. Patients were approached by healthcare professionals in the outpatient oncology clinics at Juan Ramón Jiménez University Hospital, who informed them about the study and provided a brochure with all the details and various ways to participate: via a web link/QR code to complete the form independently or by indicating if they wished to be contacted by phone. Participants were contacted by phone, after obtaining their informed consent, to remind them to complete the form or to complete it over the phone. The estimated time for completing all data was approximately 25 min. Three other tools were used in order to assess divergent/convergent validity as described:Resilience Scale (RS-14): Measures individual resilience, considered a positive personality trait that allows adaptation to adverse situations. Originally developed by Wagnild [19] and revised in 2009 to the current 14-item scale. Adapted to the Spanish version by Sánchez-Teruel [20] in 2015. The RS-14 measures two factors: personal competence (11 items including self-confidence, independence, decision-making, ingenuity, and perseverance) and acceptance of self and life (3 items—adaptability, balance, flexibility, and a stable life perspective). The author of the original scale defines the following levels of resilience: 98–82 = very high resilience; 81–64 = high resilience; 63–49 = normal; 48–31 = low; 30–14 = very low.Coping Strategies Inventory (CSI): Measures the frequency of use of coping strategies. Validated and structured by Tobin [21] and in its Spanish version of 40 items adapted by Cano [22]. The person starts by describing the stressful situation and responds on a five-point Likert scale that defines the frequency of what action was taken in the described situation. An item regarding perceived self-efficacy in coping is added at the end. The inventory is structured around eight coping strategies: problem-solving (strategies aimed at eliminating stress by modifying the situation), self-criticism (self-blame for the emergence of the situation and apparent mismanagement), emotional expression (releasing emotions resulting from stress), wishful thinking (desire for the situation to be less stressful), social support (seeking emotional support), cognitive restructuring (modifying the meaning of the situation), problem avoidance (denial and avoidance of thoughts and actions related to the situation), and social withdrawal (withdrawing from close people related to the stressful situation).SF-36 Health Survey: A generic health profile scale commonly used as a quality-of-life assessment tool. Developed by McHorney [23] and adapted to the Spanish version by Alonso [24], showing adequate levels of validity, reliability, and cultural equivalence with versions developed in other countries. It consists of 36 items and 8 dimensions: physical functioning (PF), role limitations due to physical problems (RP), bodily pain (BP), general health (GH), vitality (VT), social functioning (SF), role limitations due to emotional problems (RE), and mental health (MH). In addition to these eight health scales, there is an item that evaluates the general concept of changes in the perception of current health status compared to how it was one year ago (CS). For each dimension, the items are coded, summed, and transformed into a scale ranging from 0 (worst health status) to 100 (best health status).

#### 2.2.6. Analysis and Interpretation of the Data

In an initial descriptive analysis, categorical variables were expressed as percentages and frequencies, while continuous variables were presented as means, standard deviations, and minimum–maximum values. Skewness and kurtosis values were calculated for each of the items.

Construct validity through confirmatory factor analysis

A confirmatory factor analysis (CFA) was performed based on the unidimensional models initially proposed for the Fear of Cancer Recurrence scale (FCR4 and FCR7). The adequacy of the data for CFA was assessed using the Kaiser–Meyer–Olkin (KMO) index and Bartlett’s test of sphericity. Values greater than 0.75 for the KMO and statistically significant values of *p* ≤ 0.05 for Bartlett’s test were considered adequate [17,25]. The unidimensionality of the model was evaluated using the Unidimensional Congruence (UniCo), Explained Common Variance (ECV), and Mean of Item Residual Absolute Loadings (MIREAL) indices. UniCo values greater than 0.95, ECV values greater than 0.85, and MIREAL values below 0.30 were considered indicative that the data could be regarded as essentially unidimensional [26].

According to the analysis of skewness and kurtosis of the items, a polychoric correlation matrix was used for the factor analysis (FA), with factor extraction performed using Robust Unweighted Least Squares (RULS) and oblique PROMIN rotation [17,25]. Parallel analysis was used to determine the number of factors to retain, and the consistency of the retained factors was calculated. Confidence intervals at 95% for item scores and model measures were computed using bootstrapping. To assess the adequacy of the factor solution, the following indices were used: Root Mean Square of Residuals (RMSR), Root Mean Square Error of Approximation (RMSEA), Non-Normed Fit Index (NNFI), Comparative Fit Index (CFI), Goodness-of-Fit Index (GFI), and Adjusted Goodness-of-Fit Index (AGFI). For RMSEA, values below 0.05 were considered a good fit, and values between 0.05 and 0.08 indicated a reasonable fit [17,25]. NNFI and CFI values of 0.95 or higher, and GFI and AGFI values above 0.90, were regarded as indicators of a good model fit. For RMSR, Kelley’s criterion was used, where the RMSR value is compared to the typical error that a correlation of 0 would have in the population [17].

The consistency of the factor was evaluated using the ORION coefficients (overall reliability of fully-informative prior oblique N-EAP scores) and the Factor Determinacy Index (FDI) [27]. Additionally, the sensitivity ratio (SR) and the expected percentage of true differences (EPTD) were used. SR can be interpreted as the number of distinct factor levels that can be differentiated based on the factor score estimates. The expected percentage of true differences (EPTD) refers to the estimated percentage of differences between the observed factor score estimates that are in the same direction as the corresponding true differences. If factor scores are intended for individual assessment, FDI values above 0.90, marginal reliabilities above 0.80, SR values above 2, and EPTDs above 90% are recommended [26]. The Generalized G-H index (H-latent and H-observed) was calculated to assess the extent to which the items reflected a common factor. H-latent evaluates how well the factor can be identified by the continuous latent response variables underlying the observed item scores, while H-observed evaluates how well the factor can be identified from the observed item scores. Values above 0.80 are considered indicators of a well-defined latent variable, which is more likely to remain stable across studies, whereas low values suggest a poorly defined latent variable, which is likely to change between studies [28].

Construct-structural validity through Rasch analysis

Given the unidimensionality of the model, a Rasch analysis was conducted. The assumption of local independence among items was tested using Yen’s Q3 test [29]. The parameters were estimated using the Joint Maximum Likelihood Estimation (JMLE) method for an Andrich’s Rating Scale Model. The fit of the items and persons was estimated using the outfit (unweighted mean square fit statistic, UMS) and infit (weighted mean square fit statistic, WMS), as well as the standardized unweighted mean square fit statistic (Std. UMS) and standardized weighted mean square fit statistic (Std. WMS). For UMS and WMS, fit indices between 0.8 and 1.2 were considered indicative of a good fit, and values between 0.5 and 1.5 were considered acceptable [30]. Std. UMS and Std. WMS values greater than 3 indicated highly unexpected data for the model, while values between −1.9 and 1.9 reflected reasonably predictable data [31]. Quality statistics (separation indices and reliability) were calculated for both the items and the persons. For persons, reliability values above 0.8 and separation values greater than 2 were considered desirable.

Reliability

Reliability (internal consistency) was assessed using the omega and alpha coefficients, with their respective 95% confidence intervals calculated for both coefficients.

Divergent/Convergent Validity

To explore potential positive–negative correlations between FCR7-SP and three other instruments (RS-14, CSI and SF-36), Spearman’s correlation coefficient was used, as the normality test (Kolmogorov–Smirnov) determined a non-symmetric distribution of FCR7 scores. A negative correlation was considered when the correlation coefficient (r) was greater than −0.5.

Known-groups validation

A bivariate inferential analysis was conducted to compare the models obtained for the scale (known-groups validation). After verifying the asymmetry of the data distribution using the Kolmogorov–Smirnov test, the non-parametric Mann–Whitney U test was used to compare means between two groups, and the Kruskal–Wallis test was used for comparisons between more than two groups. A post hoc test (Dwass–Steel–Critchlow–Fligne) was conducted to identify which groups showed differences. A statistical significance level of α ≤ 0.05 was set for this study. For each association studied, the effect size was calculated using Hedges’ g formula and Kelley’s epsilon-squared measure. The descriptive and inferential analysis of the studied variables was performed using the statistical package JAMOVI©v.2.3.24. The factor analysis and model reliability analysis were performed with the free access software FACTOR© Release Version 12.02.01 × 64 bits, and the Rasch analysis was conducted using JMetrik© Software Version 2.0.

#### 2.2.7. Ethics Considerations

The applicable ethical principles in healthcare research outlined in the Declaration of Helsinki and the Belmont Report were followed. Additionally, the provisions of Spanish Organic Law 7/2021, of 26 May, on the protection of personal data processed for the purposes of prevention, detection, investigation, and prosecution of criminal offenses and the execution of criminal penalties were adhered to. The protection of privacy, confidentiality, and data security was ensured. An information sheet was provided, and informed consent was obtained from each participant. This study received a favorable opinion from the Provincial Research Ethics Committee of Huelva and the management of the participating center, PI 2167/N/21.

## 3. Results

### 3.1. Stage 1: Translation Procedure

The translators indicated a low level of complexity in the language used. In Item 1, the expression “I fear that cancer may appear” was agreed upon instead of “I am scared that cancer will come back”. In Item 4, regarding the word “fluctuations”, the appropriateness of using one word or another was discussed, reaching a consensus on this concept due to cultural relevance. Nuances such as whether to ask in the impersonal or first person were also debated, and it was decided to follow the recommendation of the author of the original questionnaire. Following this, the first version of the FCR7-SP scale was obtained.

#### 3.1.1. Content Validity Results

After evaluation by the expert group, all items received Aiken’s V coefficient values greater than 0.70. Table 1 shows the scores assigned by each expert for each item, along with the values obtained and their respective 95% confidence intervals.

#### 3.1.2. Face Validity Results

The results of the face validity assessment highlighted some issues in the wording and acceptability of certain items. Minor changes were made to the wording of Items 4 and 7 to improve comprehension. Eight participants (40%) suggested modifying the response format for Item 7 to use a scale of 1 to 5, like the other items, instead of the original 1 to 10 scale, resulting in a total FCR7-SP score ranging from 7 to 35 points. The score of 69.42 on the INFLESZ scale indicated a “fairly easy” level of comprehensibility for the FCR7-SP scale.

### 3.2. Stage 2: Descriptive Analysis of the Sample and the Items of the Fear of Recurrence-FCR7-SP

A total of 315 participants were recruited (n = 315), comprising 268 women (85.1%) and 47 men (14.9%), with a mean age of 57.91 years (SD = 11.80) (minimum age 29, maximum age 90). Of the participants, 94.0% were Spanish, while only 6.0% were from other countries. Regarding marital status, 42.2% (175) were married, 14.6% (46) were divorced, 10.2% (32) were widowed, 9.2% (29) were single, 6.7% (21) had partners, and 3.5% (11) were separated. In terms of education level, 21.6% (n = 68) of participants had a university education, 21.0% (n = 66) had a secondary education, and 57.5% (181) had a primary education or no formal education.

The distribution by cancer type shows that 63.5% (200) of patients have a diagnosis of breast cancer, 20.0% (63) have colorectal cancer, and 16.5% (52) have other types. At the time of diagnosis, 40.7% (169) of participants were in stage II, 17.1% (71) were in stage I, 9.9% (41) were in stage III, and 2.2% (9) were in stage IV. Regarding the treatment received, 27.6% (87) underwent surgery and chemotherapy, 24.1% (76) had surgery and radiotherapy, 23.2% (73) had only surgical intervention, and 19.4% (61) received chemotherapy and radiotherapy in addition to surgery. As to whether they have sought psycho-oncological care, 94.6% (298) responded no, and 5.4% (7) responded yes.

The descriptive analysis of the items (means and their confidence intervals, standard deviation, and floor and ceiling scores), as well as the values of skewness and kurtosis, can be found in Table 2.

#### 3.2.1. Construct Validity Through Confirmatory Factor Analysis for FCR4-SP and FCR7-SP

A confirmatory factor analysis (CFA) was conducted based on the initial unidimensional models proposed for FCR4 and FCR7. For the FCR4-SP model, the KMO values and Bartlett’s test statistic indicated sufficient sample adequacy (KMO = 0.788 [95% CI: 0.693–0.837]; Bartlett’s *p* ≤ 0.001). The eigenvalue for the first factor was 3.555, and 0.217 for the second, providing a unidimensional solution with an explained variance of 91.01% (parallel analysis recommended a one-factor solution). The fit values for this model were RMSEA = 0.129 [95% CI: 0.115–0.141], NNFI = 0.977 [95% CI: 0.975–0.979], CFI = 0.992 [95% CI: 0.992–0.993], GFI = 1.000 [95% CI: 0.999–1.000], and AGFI = 0.999 [95% CI: 0.997–1.000], indicating an acceptable model fit. The RMSR was 0.021 [95% CI: 0.012–0.032] (the expected RMSR value according to Kelley’s criterion for an acceptable model in this case was 0.056). In this model, all items had loadings above 0.850.

For the FCR7-SP model, the KMO values and Bartlett’s test statistic indicated better sample adequacy (KMO = 0.802 [95% CI: 0.382–0.869]; Bartlett’s *p* ≤ 0.001). The SENECA estimation recommended a minimum sample size of 230 subjects for a non-linear model (polychoric matrix). The eigenvalue for the first factor was 5.601, and 0.733 for the second, providing a unidimensional solution with an explained variance of 81.31% (parallel analysis also recommended a one-factor solution). The unidimensionality analysis yielded the following results: UniCo = 0.953 [95% CI: 0.914–0.982], ECV = 0.886 [95% CI: 0.853–0.920], MIREAL = 0.211 [95% CI: 0.174–0.230], indicating that the instrument can be considered essentially unidimensional. The fit values for this model were RMSEA = 0.117 [95% CI: 0.110–0.124], NNFI = 0.977 [95% CI: 0.974–0.979], CFI = 0.985 [95% CI: 0.983–0.986], GFI = 0.994 [95% CI: 0.989–0.997], and AGFI = 0.991 [95% CI: 0.983–0.995], indicating an acceptable model fit. The RMSR was 0.067 [95% CI: 0.049–0.084] (the expected RMSR value according to Kelley’s criterion for an acceptable model in this case was 0.056). Table 3 displays the factor loadings (after rotation) for the two models (FCR4-SP and FCR7-SP).

In this model, all items had loadings above 0.800, except for Item 6: “I examine myself to see if I have physical signs of cancer”. This item had loadings above 0.500.

Finally, regarding the replicability of the construct, the H-latent value obtained for the FCR4-SP model was 0.973 [95% CI: 0.960–0.985] and H-observed was 0.874 [95% CI: 0.855–0.887]. For the FCR7-SP model, the H-latent value for the factor was 0.972 [95% CI: 0.962–0.978] and H-observed was 0.909 [95% CI: 0.896–0.935], indicating a well-defined latent variable in both cases.

#### 3.2.2. Construct-Structural Validity Through Rasch Analysis

As the factor analysis clearly suggested a unidimensional solution, similar to the study by Humphris et al. [13], a Rasch analysis was conducted to evaluate the construct validity of the scale. For this, the scale scores were reconverted, transforming the item scores (from 0 to 4 points). Regarding the necessary assumption of local independence (Yen Q3 test), the correlation matrix showed the majority of values below 0.2–0.3, indicating that the local independence of the items was valid.

In the FCR7-SP model, Items 4, 5, and 7 showed infit (WMS) and outfit (UMS) values with a good fit, while Items 1, 2, and 3 had an acceptable fit. Item 6 initially exhibited a poor fit on both indices (WMS of 2.44 and UMS of 11.58). Due to this poor fit of Item 6, the analysis was repeated without this item, testing a model referred to as FCR6-SP (without Item 6). Removing Item 6 resulted in an improvement in model fit, with better infit and outfit values for most items, except for Items 1 and 5, which still had a poor fit on the outfit index. Additionally, the fit of the FCR4-SP model was tested, where only Items 1 and 3 showed a good fit (Items 2 and 4 had an acceptable fit). Table 4 presents the infit (WMS) and outfit (UMS) values obtained for the various models: the FCR7-SP model (which includes all seven items), the FCR6-SP model (where Item 6 was removed), and the FCR4-SP model (which includes Items 1 to 4).

#### 3.2.3. Reliability

The overall omega and Cronbach’s alpha coefficients were 0.933 [95% CI: 0.922–0.944] and 0.926 [95% CI: 0.926–0.937], respectively, for the FCR7-SP model. The reliability analysis indicated that the removal of Item 6 led to an increase in the internal consistency reliability values of the scale (Table 6). For the FCR6-SP model, the values were 0.943 [95% CI: 0.933–0.953] for the omega coefficient and 0.942 [95% CI: 0.931–0.951] for Cronbach’s alpha coefficient.

#### 3.2.4. Divergent–Convergent Validity

The mean total score for the 14-Item Resilience Scale (RS-14) was 71.68 (SD = 15.96) points. A Spearman correlation coefficient of r = −0.179 (*p* < 0.001) indicated a negative correlation between the FCR7-SP scale score and the total score of RS-14. For the FCR6-SP model, the Spearman correlation coefficient was r = −0.178 (*p* < 0.001).

The mean total score for the Coping Strategies Inventory (CSI-40) was 87.60 (SD = 13.34) points. A Spearman correlation coefficient of r = −0.230 (*p* < 0.001) indicated a negative correlation between the FCR7-SP scale score and the total score of the CSI-40. For the FCR6-SP model, the Spearman correlation coefficient was r = −0.220 (*p* < 0.001).

In this regard, there is also a negative correlation with the general health and mental health dimensions of SF-36 for both models (Table 7).

#### 3.2.5. Validation by Known Groups and Final Proposal

The mean total score was 26.5 (SD = 6.49) for the FCR7-SP model and 22.0 (SD = 5.98) for the FCR6-SP model. For the validation by known groups, the association between several variables and the total score of the scale was analyzed for both the FCR7-SP and FCR6-SP models, also measuring the effect sizes. Statistically significant differences were found concerning gender, nationality, and active treatment in both models, with medium effect sizes for nationality and large effect sizes for gender and active treatment (Table 8).

Regarding the relationship between the fear of recurrence and cancer type, there are statistically significant differences among the different types of cancer (breast cancer, colon cancer, and other types) for both models (FCR7-SP and FCR6-SP) (Table 9).

As shown in Table 9, the highest levels of fear of recurrence (FCR) obtained in both models correspond to breast cancer, followed by colon cancer, and thirdly, other types of cancer.

## 4. Discussion

FCR7 is a versatile tool for application in clinical practice, which has seen widespread international expansion in recent years with the emergence of various translated and adapted versions in different languages: Chinese, Tamil, and Portuguese. Additionally, abbreviated versions with fewer items have been developed [32,33,34].

As mentioned earlier, there is a shortage of tools to assess fear of recurrence in Spanish, making it necessary to conduct the transcultural adaptation and validation of this tool for use with cancer patients of various types. In fact, some previous studies conducted in Spain on this topic, such as the multicenter study by Calderón [35], resort to another tool (Cancer Worry Scale) to measure the fear of recurrence. Although this tool is validated, it was designed for use in the general healthy population (without a history of cancer) due to the lack of specific instruments for cancer patients.

Regarding the content validation process, the panel of 10 experts showed positive results, with all items achieving acceptable values of Aiken’s V coefficient. However, for Items 4, 5, and 6, the lower confidence intervals fell below the value of 0.70 recommended by Charter [36]. Interestingly, Item 6, which later analyses indicated issues with, was not the item with the lowest value (v = 0.83 [95% CI: 0.66–0.93]); rather, it was Item 5, which had a value of 0.77 [95% CI: 0.59–0.88].

After minor adjustments were made following the pilot testing, the wording of Items 4 and 7 improved the questionnaire’s comprehension and acceptability. Additionally, it was deemed necessary to change the measurement system for Item 7 (from a scale of 1–10 to a scale of 1–5) to standardize the scoring system with the rest of the questionnaire, facilitating patient assessment and the validation process, as recommended for questionnaire design [37,38]. Concerning the sample size, both the KMO and SENECA estimates indicated sufficient sample adequacy. The composition reflects a typical distribution found in oncology studies, where breast cancer is often the most prevalent.

The confirmatory factor analysis (CFA) clearly demonstrated a unidimensional solution for FCR7-SP, similar to other validation studies [32,33,34]. The fit indices for the FCR7-SP model were acceptable, although the RMSEA showed a high value (RMSEA = 0.117 [95% CI: 0.110–0.124]), which may suggest a moderate fit for the model. This same finding was reported by Bergerot et al. [34] in the Brazilian validation, with RMSEA values of 0.099 [95% CI: 0.063–0.138], but adequate CFI values (0.971 [95% CI: 0.099–0.063], similar to our study).

The factorial loadings of the items are high, above 0.800 (except for Item 6), along with the obtained h-latent value (0.972), indicating that the instrument consistently measures cancer recurrence fear. Given that the unidimensional structure of the scale appears confirmed, and the analysis with the Yen Q3 test supported local independence, a Rasch analysis was conducted. While the original study with the FCR7 [13] also performed an Item Response Theory (IRT) analysis, the rest of the validation studies did not use this approach. In our case, this analysis indicated poor functioning of Item 6, suggesting its removal and the proposal of an alternative model for the Spanish version with 6 items (FCR6-SP). The fit values, reliability (omega and Cronbach’s alpha), and separation indices slightly improved for this newly proposed model (FCR6-SP).

In the original study by Humphris [13], the lowest loading was also obtained for this item (0.52). Similarly, in the validation study of the Chinese version [32], this item had the lowest factorial loading (0.578), as also seen in the study by Bergerot et al. [34] (factorial loading of 0.469). The study by Nandakumar [33] indicated that all items had a significantly strong correlation with each other except for Item 6 (0.616), and removing this item improved the Cronbach’s alpha coefficient. Finally, these results align with data from the study by Braun [39], which focused on validating a version of the instrument for patients with brain tumors (FCR6-Brain). In that study, the removal of Item 6 also improved the metrics of the single factor and the convergent validity, leading the authors to favor a model similar to FCR6-SP with six items instead of seven.

A possible explanation for the behavior of this item may be that it refers to physical self-examination, which contrasts with the other items that address more emotional perceptions. Additionally, this item might be influenced by the type of cancer, as the significance of self-examination varies across different cancers. However, in the analysis by known groups, there were no differences regarding the application of the two models and the type of cancer. Perhaps the most sensible approach when choosing between models is to evaluate the target population and determine the potential importance of Item 6, particularly in relation to the type of cancer. This aspect, nonetheless, should be explored more thoroughly in future research.

Regarding the FCR4-SP model, the results provide evidence of adequate content validity, similar to that found by Bergerot [34] in their conceptual adaptation with 200 cancer patients. However, the FCR4-SP model does not introduce significant changes compared to the initial FCR7-SP model, unlike the study conducted by Humphris [13] with 256 breast and colon cancer patients, which demonstrated more substantial modifications. In relation to the validation process of the various translated versions of the instrument, there is considerable heterogeneity, showcasing the versatility of the questionnaire for use in patients with different types of cancer. The Chinese version by Yang [32] was validated in a group of patients with various cancers, while the Tamil version by Nandakumar [33] and the Brazilian-Portuguese version by Bergerot [34] focus on breast cancer patients. The FCR6-Brain version by Braun [39] is applied to patients with brain tumors. In all cases, the content of the original items has been maintained, achieving adequate reliability indices, but the previously mentioned issues regarding Item 6 persist.

This passage discusses the work by Iglesias-Puzas et al. [40] regarding a Spanish version of the FCR7 instrument, which was specifically adapted for skin cancer patients. It highlights that this initial validation involved 123 patients with non-metastatic melanoma and emphasizes the novelty of the initiative. However, it points out that potential issues with Item 6 were not identified, as the study did not report factor loadings or model fit indices, focusing solely on reliability measures like Cronbach’s alpha and stability assessed by intraclass correlation. The text suggests that comparisons with FCR7-SP and FCR6-SP should be approached cautiously due to the specific adaptations and the limited population studied. Despite these differences, it notes that the reliability and construct validity values are similar across both Spanish versions, indicating that both the FCR7-SP and FCR6-SP can be applied to various cancer types while maintaining the original questionnaire’s generic nature.

Like the other studies, the correlation analysis in this case has shown consistent results, establishing negative correlations between the FCR6/7-SP and the resilience scales (RS-14), quality of life (SF-36) (in all dimensions—physical functioning, physical role, body pain, general health, vitality, emotional role, and mental health—except in social functioning), and the Coping Strategies Inventory (CSI). This suggests that greater fear of recurrence is associated with lower resilience, poorer coping, and lower quality of life, and conversely, greater resilience and coping ability correlate with less fear of recurrence and better quality of life. These results are consistent with the existing literature [41,42].

Among the possible limitations of the study, it is noted that the questionnaire was administered in two modalities: self-administered and interviewer-administered, which may have influenced the interpretation of the items. However, this was minimized in cases where the survey was conducted over the phone, limiting the reading of the items without any conditioning in the response.

Furthermore, although the questionnaire was administered to oncology patients with different types of cancer, most of the sample consisted of breast cancer patients, which may have influenced the evaluation of the various items, although the composition of the sample reflects the typical distribution in the population according to the prevalence of different types of cancer.

Being aware of the potential effect of the sample composition, we have maintained the initial orientation of the original tool which has been designed to be applied in any type of oncology patient. Nevertheless, it would be interesting to continue the psychometric analysis and to investigate the effect of the use of the tool on the different types of cancer separately. In this way, it would be possible to determine which of the two models (FCR6-SP or FCR7-SP) should be used for each type of cancer. In this initial validation study, we have observed that Item 6 (related to self-examination) can behave differently according to the type of cancer, as self-examination is especially indicated in some types of cancer, while in others it has less clinical relevance, justifying different results in each case. However, it will be pertinent to continue the analysis of the tool properties using different types of cancer in future research.

Finally, although the original tool does not include cut-off points, in order to enhance its clinical applicability and facilitate its interpretation, it would be of interest to establish cut-off points based on different levels of fear, an aspect that has already being considered in ongoing studies.

## 5. Conclusions

After the translation and cultural adaptation process of the original “Fear of Cancer Recurrence” (FCR) tool, the Spanish version FCR-SP has been obtained, which can be applied based on three models: FCR7-SP (which includes the seven original items), FCR6-SP (where Item 6 is removed), and FCR4-SP. However, analyses support the use of FCR6/7-SP, as the FCR4-SP model does not provide substantial improvements in terms of its validity and time savings for administering the instrument.

These two proposed models, FCR6/7-SP, are valid, demonstrating adequate content validity, face validity, construct validity, and reliability, with the FCR6-SP model showing slightly better values. Therefore, it is considered that after the transcultural adaptation and validation process, the FCR6-SP version offers the greatest validity and reliability and constitutes a highly useful tool for use in the Spanish-speaking context, although the application of the FCR7-SP model may be valued in cases where the type of cancer may influence the importance of the subject’s self-examination.

## Figures and Tables

**Table 1 cancers-17-00875-t001:** Expert scores and content validation with Aiken’s V coefficient.

FCR7-SP Items	Exp 1	Exp 2	Exp 3	Exp 4	Exp 5	Exp 6	Exp 7	Exp 8	Exp 9	Exp 10	V Aiken Coefficient [95% CI]
1. I’m afraid cancer may appear	4	4	3	4	4	3	4	4	2	4	0.87 [0.70–0.95]
2. I am worried or anxious about the possibility of cancer recurrence	4	4	3	4	4	3	3	4	3	4	0.87 [0.70–0.95]
3. How often do you worry about getting cancer again?	4	4	3	4	3	3	4	4	4	4	0.90 [0.74–0.97]
4. I have strong feelings about the cancer coming back	4	4	3	4	3	3	3	4	3	4	0.83 [0.66–0.93]
5. I think about the cancer coming back when I don’t want to	4	4	3	4	4	3	1	4	2	4	0.77 [0.59–0.88]
6. I check myself for physical signs of cancer	4	4	4	3	3	4	4	3	3	3	0.83 [0.66–0.93]
7. To what extent does worry about developing cancer again take over or interfere with your thoughts and activities?	4	4	2	4	4	4	2	4	4	4	0.87 [0.70–0.95]

Exp: Expert.

**Table 2 cancers-17-00875-t002:** Descriptive analysis of the FCR7-SP items (means and their confidence intervals, standard deviation, and floor and ceiling scores).

Items FCR7-SP	M [95%CI] ^1^	SD ^2^	Symmetry	Kurtosis	Floor Not at All ^3^ n (%)	Ceiling All the Time ^3^ n (%)
Item 1. I am afraid that cancer may appear.	4.25 [4.15–4.36]	0.98	−1.227	0.933	6 (1.9%)	172 (54.6%)
Item 2. I am worried or anxious about the possibility of cancer recurrence.	3.91 [3.78–4.03]	1.12	−0.841	0.050	15 (4.8%)	124 (39.4%)
Item 3. How often do you worry about getting cancer again?	3.85 [3.73–3.96]	1.05	−0.636	−0.263	8 (2.5%)	104 (33.0%)
Item 4. I have strong feelings about the cancer coming back.	3.26 [3.13–3.40]	1.24	−0.000	−0.997	24 (7.6%)	75 (23.8%)
Item 5. I think about the cancer coming back when I don’t want to.	2.93 [2.78–3.08]	1.36	0.178	−1.110	56 (17.8%)	62 (19.7%)
Item 6. I check myself for physical signs of cancer.	4.54 [4.43–4.65]	1.00	−2.305	4.485	12 (3.8%)	246 (78.1%)
Item 7. To what extent does worry about developing cancer again take over or interfere with your thoughts and activities?	3.80 [3.69–3.91]	1.00	−0.626	0.168	10 (3.2%)	88 (27.9%)

^1^ Mean [95% confidence interval]; ^2^ standard deviation; ^3^ only the highest (ceiling) and lowest scores (floor) per question are shown.

**Table 3 cancers-17-00875-t003:** Factor loadings (after rotation) of the FCR7-SP and FCR4-SP models, with quality and reliability estimators and fit values for the two models.

Items	FCR4-SP Factor Loading Values [95% Confidence Interval]	FCR7-SP Factor Loading Values [95% Confidence Interval]
Item 1. I am afraid that cancer may appear.	0.903 [0.858–0.935]	0.913 [0.853–0.941]
Item 2. I am worried or anxious about the possibility of cancer recurrence.	0.976 [0.951–0.992]	0.951 [0.926–0.972]
Item 3. How often do you worry about getting cancer again?	0.944 [0.915–0.963]	0.921 [0.885–0.943]
Item 4. I have strong feelings about the cancer coming back.	0.869 [0.822–0.902]	0.904 [0.859–0.932]
Item 5. I think about the cancer coming back when I don’t want to.		0.869 [0.817–0.905]
Item 6. I check myself for physical signs of cancer.		0.624 [0.517–0.727]
Item 7. To what extent does worry about developing cancer again take over or interfere with your thoughts and activities?		0.925 [0.869–0.950]
Factor Determinacy Index (FDI)	0.986	0.986
EAP score reliability	0.973	0.972
Sensitivity ratio (SR) ^a^	5.992	5.857
Expected percentage of true differences (EPTD) ^b^	97.1%	97.0%
**Coefficients**	**FCR4-SP Model Fit Values** **[95% Confid. Interval]**	**FCR7-SP Model Fit Values** **[95% Confidence Interval]**
Root Mean Square Error of Approximation (RMSEA)	0.129 [0.115–0.141]	0.117 [0.110–0.124]
Non-Normed Fit Index (NNFI)	0.977 [0.975–0.979]	0.977 [0.974–0.979]
Comparative Fit Index (CFI)	0.992 [0.992–0.993]	0.985 [0.983–0.986]
Goodness-of-Fit Index (GFI)	1.000 [0.999–1.000]	0.994 [0.989–0.997]
Adjusted Goodness-of-Fit Index (AGFI)	0.999 [0.997–1.000]	0.991 [0.983–0.995]
Root Mean Square of Residuals (RMSR)	0.021 [0.012–0.032]	0.067 [0.049–0.084]

^a^ = The sensitivity ratio (SR) can be interpreted as the number of different factor levels than can be differentiated on the basis of the factor score estimates. ^b^ = Expected percentage of true differences (EPTD) is the estimated percentage of differences between the observed factor score estimates that are in the same direction as the corresponding true differences. If factor scores are to be used for individual assessment, FDI values above 0.90, marginal reliabilities above 0.80, SR above 2, and EPTDs above 90% are recommended.

**Table 4 cancers-17-00875-t004:** Rasch analysis results for the FCR7-SP, FCR6-SP, and FCR4-SP models.

Item	Index of Difficulty * FCR7-SP	Infit WMS ** (Std. WMS ***) FCR7-SP	Outfit UMS ** (Std. UMS ***) FCR7-SP	Infit WMS ** (Std. WMS ***) Without Item 6-Model FCR6-SP	Outfit UMS ** (Std. UMS ***) Without Item 6-Model FCR6-SP	Infit WMS ** (Std. WMS ***) Model FCR4-SP	Outfit UMS ** (Std. UMS ***) Model FCR4-SP
Item 1. I am afraid that cancer may appear.	−1.40	0.74 (−3.03)	1.22 (1.28)	**0.93** (−0.72)	1.74 (3.06)	**1.08** (**0.80**)	1.33 (**1.52**)
Item 2. I am worried or anxious about the possibility of cancer recurrence.	−0.29	0.66 (−4.21)	0.72 (−2.92)	**0.80** (−2.36)	**0.86** (**−1.31**)	0.70 (−3.68)	0.66 (−3.73)
Item 3. How often do you worry about getting cancer again?	−0.31	0.78 (−2.73)	0.79 (−2.57)	**0.97** (−0.39)	**0.95** (**−0.56**)	**0.87** (−1.60)	**0.85** (−1.66)
Item 4. I have strong feelings about the cancer coming back.	1.39	**0.87** (−1.60)	**0.87** (−1.56)	**0.86** (−1.71)	**0.87** (**−1.45**)	1.33 (3.29)	1.31 (2.81)
Item 5. I think about the cancer coming back when I don’t want to.	2.47	**1.10** (1.10)	**1.12** (1.16)	**1.11** (1.25)	2.12 (4.79)	NA	NA
Item 6. I check myself for physical signs of cancer.	−1.74	2.44 (7.83)	11.58 (7.10)	NA	NA	NA	NA
Item 7. To what extent does worry about developing cancer again take over or interfere with your thoughts and activities?	−0.13	**0.81** (−2.28)	**0.84** (1.91)	**1.18** (2.06)	**1.19** (1.97)	NA	NA

* Difficulty index: In this case, it indicates the highest values of fear of cancer recurrence. Calculated for the seven items. ** Outfit unweighted mean square fit statistic (UMS) and infit weighted mean square fit statistic (WMS): Values of the fit indices between 0.8 and 1.2 mean a good fit, and values between 0.5 and 1.5 mean an acceptable fit (productive for the measurement). Values >2.0 distort or degrade the measurement system. *** Std. UMS: standardized value (Std) unweighted mean square fit statistic (UMS); Std. WMS: standardized value infit weighted mean square fit statistic (WMS): values ≥3 indicate data that are very unexpected if they fit the model (perfectly), so they probably do not/values 2.0–2.9 indicate data that are noticeably unpredictable/values −1.9–1.9 indicate data that have reasonable predictability/values ≤−2 indicate data that are too predictable. Other “dimensions” may be constraining the response patterns. NA, not applicable. Regarding the quality statistics of the scale, Table 5 provides the reliability values for the items and individuals across the three analysis models. Although the values are very similar, they were slightly better for the FCR6-SP model. In all cases, the separation index for both individuals and items exceeded 2.

**Table 5 cancers-17-00875-t005:** Comparison of reliability values according to the Rasch analysis of the items and the people in the analysis models.

	Model FCR7-SP (Seven Items)		Model FCR6-SP (Item Number 6 Eliminated)		Model FCR4-SP (Item Number 1 to Number 4)	
	Items	Persons	Items	Persons	Item	Persons
Separation index	13.181	3.153	14.777	3.723	13.212	3.164
Reliability	0.994	0.908	0.995	0.932	0.994	0.909

**Table 6 cancers-17-00875-t006:** Reliability analysis through the omega coefficient and Cronbach’s alpha.

	If Item Dropped	
Item	Cronbach’s α	McDonald’s ω
FCR1	0.912	0.92
FCR2	0.904	0.912
FCR3	0.906	0.915
FCR4	0.908	0.917
FCR5	0.916	0.922
FCR6	0.942	0.945
FCR7	0.908	0.917

**Table 7 cancers-17-00875-t007:** Correlations between the scores of the FCR7-SP model and the FCR6-SP model and the different dimensions of SF-36. * Statistically significant *p*-value Spearman’s Rho.

SF-36 Dimensions Scores				
	FCR7-SP		FCR6-SP	
		*p*-Value		*p*-Value
Physical functioning	−0.132	0.019 *	−0.147	0.009 *
Physical role	−0.176	0.002 *	−0.183	0.001 *
Body pain	−0.293	≤0.001 *	−0.295	≤0.001 *
General Health	−0.404	≤0.001 *	−0.414	≤0.001 *
Vitality	−0.365	≤0.001 *	−0.376	≤0.001 *
Social Functioning	0.084	0.138	0.104	0.066
Emotional Role	−0.391	≤0.001 *	−0.406	≤0.001 *
Mental Health	−0.451	≤0.001 *	−0.464	≤0.001 *

**Table 8 cancers-17-00875-t008:** Validation by known groups using bivariate analysis for both models.

	Model FCR 7-SP	Model FCR6-SP
	M (SD) ^1^	M (SD) ^1^
**Gender**		
Female n = 289	27.3 (6.19)	22.8 (5.65)
Male n = 47	22.2 (6.51)	17.7 (6.11)
*p*-value ^2^	<0.001 *	<0.001 *
Effect size ^3^ (Hedges’ g)	0.816	0.890
**Children**		
If n = 257	26.4 (6.48)	21.8 (5.94)
No n = 58	27.4 (6.52)	22.7 (6.15)
*p*-value ^2^	0.224	0.226
Effect size ^3^ (Hedges’ g)	0.153	0.150
**Nationality**		
Spanish n = 296	26.3 (6.54)	21.8 (6.02)
Not Spanish n = 19	30.2 (4.29)	25.5 (4.07)
*p*-value ^2^	0.013 *	0.008 *
Effect size ^3^ (Hedges’ g)	0.604	0.622
**He/she lives alone**		
Yes n = 65	26.7 (6.69)	22.1 (6.18)
Live accompanied n = 250	26.5 (6.45)	22.0 (5.94)
*p*-value ^2^	0.843	0.867
Effect size ^3^ (Hedges’ g)	0.030	0.016
**Active treatment**		
No n = 115	23.4 (6.89)	19.1 (6.36)
Yes n = 200	28.4 (5.49)	23.7 (5.06)
*p*-value ^2^	<0.001 *	<0.001 *
Effect size ^3^ (Hedges’ g)	0.826	0.824
**Psycho-oncologist during treatment**		
No n = 298	26.7 (6.37)	22.1 (5.92)
Yes n = 17	23.2 (7.84)	19.6 (6.71)
*p*-value ^2^	0.066	0.124
Effect size ^3^ (Hedges’ g)	0.541	0.418

^1^ Mean (standard deviation); ^2^ Mann–Whitney’s U-test, * statistically significant value; ^3^ effect size according to Hedges (Hedges’ g): it considers both groups’ variances and sizes. Values < 0.2 indicate small effects, 0.5 indicates a medium effect, and 0.8 indicates a large effect.

**Table 9 cancers-17-00875-t009:** Analysis by type of cancer for both models.

	Breast Cancer (a) (n = 200)	Colorectal Cancer (b) (n = 63)	Other Types of Cancer (c) (n = 52)		Kruskal–Wallis Contrast Test	Effect Size	Dwass–Steel–Critchlow–Fligne Post Hoc
	M (SD)	M (SD)	M (SD)	X ^2^	*p*-Value	Ɛ ^2^	
FCR7-SP	28.4 (5.16)	21.9 (6.57)	25.1 (7.81)	47.7	≤0.001 *	0.152	a,b (*p* = ≤0.001) ** a,c (*p* = 0.015) ** b,c (*p* = 0.040) **
FCR6-SP	23.7 (4.82)	17.6 (6.05)	20.9 (7.05)	46.6	≤0.001 *	0.148	a,b (*p* = ≤0.001) ** a,c (*p* = 0.038) ** b,c (*p* = 0.018) **

X ^2^ = Chi square. Ɛ ^2^ = Epsilon squared. Effect size from 0 to 1, with 1 being the maximum effect. * Statistically significant results according to the Kruskal–Wallis test. ** *p*-value for the two-to-two contrast.

## Data Availability

The data that support the findings of this study are available from the main author but restrictions apply as they are part of a doctoral thesis and are not publicly available yet. Data, however, will be available in the future from the authors upon reasonable request.

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
