# Peer review of "Transcultural Adaptation and Evaluation of the Psychometric Properties of the Spanish Version of the FCR7 Questionnaire for Assessing Fear of Recurrence in Cancer Patients: FCR6/7-SP"

_cancers, 2025, doi:10.3390/cancers17050875_

Round 1

Reviewer 1 Report

Comments and Suggestions for Authors

This manuscript presents a valuable contribution to the field of oncology and psycho-oncology by adapting and validating the Fear of Cancer Recurrence (FCR7) questionnaire for Spanish-speaking cancer patients. The study is well-structured, adhering to a rigorous methodological framework, including translation, cultural adaptation, and psychometric evaluation. The authors have demonstrated commendable diligence in statistical analysis and reliability assessments. However, some areas require refinement for enhanced clarity, robustness, and impact.

Major Comments:

The study appropriately justifies the removal of item 6 and reviews this in the discussion.

The study appropriately proceeds with a rigorous validation process.

The study does define cut-off points for clinical application, which improves the questionnaire’s used in practice. It would be interesting to compare FCR6/7-SP scores with clinical interviews or existing scales such as the Hospital Anxiety and Depression Scale (HADS) to further define clinically meaningful categories.

Page 6 has an unlinked reference in the following sentence: "The expected percentage of true differences (EPTD) refers to the estimated percentage of differences between the observed factor score estimates that are in the same direction as the corresponding true differences. If factor scores are intended for individual assessment, FDI values above 0.90, marginal reliabilities above 0.80, SR values above 2, and EPTDs above 90% are recommended26

The scale is evaluated for sensitivities across FCR4 factor loading and FCR7 factor loading but not over time. It would be interesting to evaluate the sensitivity of these variables in a longitudinal study measuring fear of recurrence before and after psychological interventions or follow-ups at different time points.

The study appropriately utilized acceptable values of Aiken’s V coefficient to define items which were valid.

Consider rewording the following sentence "This fear, deeply rooted in the constant and/or recurring concern of the disease reappearing..."

The term "Spanish version of FCR7" is sometimes referred to as both FCR7-SP and FCR6/7-SP. Standardize terminology throughout.

Some tables contain excessive decimal places (e.g., RMSEA=0.117 [95% CI: 0.110-0.124]). Rounding to three decimal places is sufficient for practical significance.

Consider adding graphics for the summary of model fit indices to make the comparison between FCR7-SP and FCR6-SP visually clearer.

Author Response

Please se the attachment.

Reviewer 2 Report

Comments and Suggestions for Authors

I found the paper methodologically strong assessing FCR and validating this instrument for Spanish population.

I think that the paper is good, even if I see an important limit in this paper. For this reason I think that major revisions should be made. The main limit is the huge number of women comparing to men and the predominant type of diagnosis (breast cancer).

For this reason, I suggest running the validation analyses on a unique diagnosis and gender. This could be more precise and informative. Then, with other data in the future other diagnosis could be added. This change is to be done only for the validation scores, while for the other analyses on the other instruments you can take them specifying the different sample.

For this reason, I suggest changing the title and to adjust the paper along these suggestions if it is possible for statistical reasons. If it isn’t possible, I suggest to be more cautious in the adopted terms, for example not giving this paper an objective of a Spanish validation of the scale, but the first step/a pilot study in the process of validation.

Other minor things:

Line 64: check a typing error.

It is not clear if this instrument has or not t scores and a sort of cut off measure or if it should be considered only for raw scores. Add this as a possibility in the future studies

In the discussion you could report more also the quality of life scores that are important to evidence the possible associations with the scale.

Reviewer 3 Report

Comments and Suggestions for Authors

This is a carefully conducted validation study, congratulations to the authors! My comments are mostly mild and are made to improve the quality of data presentation.

1. Lines 54 and 55. I guess the FCR is transcribed as "fear of cancer recurrence," not just "fear of recurrence."

2. Line 90. Please present the two stages of your research here.

3. Lines 94 and 95. You have already described the abbreviation "FCR7" earlier.

4. Line 177. What does "required17" stand for?

5. Subsection 2.2.5 (lines 195-247). The authors need to present the reasons for inclusion of other tools in the survey (to assess the divergent-convergent validity?).

6. Line 251. Instead of "quantitative and qualitative variables," the authors should have used more internationally common terms, i.e., continuous and categorical variables.

7. Line 311. The authors need to explain what does "Std." refer to.

8. Line 325. It is the third time when authors explain the abbreviation "FCR7".

9. Lines 394-398. The system for cancer staging used needs to be presented in the Materials and methods section.

10. Line 515. This is the fourth time the abbreviation "FCR" is explained.

11. Line 664. This is the fifth time the abbreviation "FCR" is explained.

Round 2

Reviewer 2 Report

Comments and Suggestions for Authors

The paper is really ameliorated after the revisions made by the authors following carefully the reviewers' suggestions.  

The paper has an interesting topic and now I think It could be oublished.